# The Impact of Total Deceleration Area and Fetal Growth on Neonatal Acidemia in Vacuum Extraction Deliveries

**DOI:** 10.3390/children10050776

**Published:** 2023-04-25

**Authors:** Gal Cohen, Dorit Ravid, Nagam Gnaiem, Hadar Gluska, Hanoch Schreiber, Noa Leybovitz Haleluya, Tal Biron-Shental, Michal Kovo, Ofer Markovitch

**Affiliations:** 1Department of Obstetrics and Gynecology, Meir Medical Center, Kfar Saba 4428163, Israel; 2Sackler School of Medicine, Tel Aviv University, Tel Aviv 6997801, Israel; 3Ob-Gyn Ultrasound Unit, Meir Medical Center, Kfar Saba 4428164, Israel; 4Department of Obstetrics and Gynecology, Soroka University Medical Center, Beer Sheba 8410101, Israel; sagithaleluya@gmail.com

**Keywords:** total deceleration area and vacuum extractions, umbilical cord pH in low birthweights, total deceleration area and neonatal acidemia, total deceleration area and umbilical cord pH, vacuum extraction in low birthweights

## Abstract

We aimed to investigate the correlation between total deceleration area (TDA), neonatal birthweight and neonatal acidemia in vacuum extractions (VEs). This is a retrospective study in a tertiary hospital, including VE performed due to non-reassuring fetal heart rate (NRFHR). Electronic fetal monitoring during the 120 min preceding delivery was interpreted by two obstetricians who were blinded to neonatal outcomes. TDA was calculated as the sum of the area under the curve for each deceleration. Neonatal birthweights were classified as low (<2500 g), normal (2500–3999 g) or macrosomic (>4000 g). A total of 85 VEs were analyzed. Multivariable linear regression, adjusted for gestational age, nulliparity and diabetes mellitus, revealed a negative correlation between TDA in the 60 min preceding delivery and umbilical cord pH. For every 10 K increase in TDA, the cord pH decreased by 0.02 (*p* = 0.038; 95%CI, −0.05–0.00). The use of the Ventouse-Mityvac cup was associated with a 0.08 decrease in cord pH as compared to the Kiwi OmniCup (95%CI, −0.16–0.00; *p* = 0.049). Low birthweights, compared to normal birthweights, were not associated with a change in cord pH. To conclude, a significant correlation was found between TDA during the 60 min preceding delivery and cord pH in VE performed due to NRFHR.

## 1. Introduction

Despite over half a century of using electronic fetal monitoring (EFM), as well as the introduction of the three-tiered classification system [1], considerable uncertainty remains regarding the interpretation of tracings, especially in category II. Unfortunately, these categories have not been clinically effective in predicting neonatal acidemia, even when integrated with other clinical parameters [2,3].

A prospective cohort study by Cahill et al. [2] revealed that an estimation of deceleration frequency and severity (total deceleration area (TDA)) in the second stage of labor was better than the three EFM features in predicting neonatal acidemia developed by the National Institute of Child Health and Human Development (NICHD). These findings were reconfirmed by another large cohort study that evaluated deliveries at term [4]. A recent study suggested a correlation between TDA and neonatal acidemia in cases of meconium-stained amniotic fluid [5].

Despite these promising results, concerns have been raised regarding TDA, as it is not based on timing and, therefore, does not clarify the underlying mechanism behind decelerations (hypoxic vs. non-hypoxic) [6].

Neonatal acid–base status at delivery is usually assessed using umbilical cord pH. Normal arterial cord pH values are around 7.25–7.30. [7,8] Decreased cord pH values, implying organic acid formation, are known to be associated with neonatal acidemia, though the exact threshold to define acidemia has yet to be established. Previous studies suggested cutoffs of <7.20, <7.10 and <7.00. [9] With that said, it is believed that values below 7.0 have a stronger association with neonatal morbidity and neurodevelopmental adverse outcomes [9,10,11,12].

Vacuum extraction (VE) is used in up to 13% of deliveries [13]. It is preferable to second-stage CS in cases of suspected intrapartum fetal distress because it results in quicker fetal extraction, thus lowering the risk of neonatal acidemia and hypoxic damage [14].

Obstetricians tend to prefer non-metal cups over metal ones for VE, as the latter have been linked with higher rates of birth trauma [15,16,17,18]. Some commonly used non-metal cups include the Kiwi OmniCup and the Ventouse-Mityvac vacuum-assisted delivery system. Both of these cups have a rigid, mushroom-shaped design that is disposable. The key distinctions between the two are the vacuum mechanism (handheld pump vs. conventional vacuum) and the traction pole (flexible in the Kiwi and rigid in the Mityvac). Recent data suggest that the use of the Kiwi OmniCup minimizes feto-maternal birth trauma, as compared to the Mityvac cup [19,20]. Failure rates using the Kiwi OmniCup vary [21,22,23], and are reported at around 1.4% in a previous publication from our institution [24]. Regarding the Mityvac cup, failure rates are around 0.7% [23].

Fetal growth restriction has been previously linked to neonatal acidemia [25,26].

Notably, data regarding TDA as a predictor of neonatal acidemia in the scenario of VE are sparse, as well as evidence regarding the possible effect of neonatal birthweight on the risk for acidemia during VE. 

Our study aimed to investigate the correlation between TDA and neonatal cord pH in VE performed for the indication of non-reassuring fetal heart rate (NRFHR). We also aimed to evaluate the correlation between neonatal birthweight and cord pH in that scenario. 

## 2. Materials and Methods

This retrospective cohort study included women who underwent VE, performed in a single tertiary care medical center at gestational age (GA) > 34 weeks for the indication of NRFHR between 1 January 2015 and 1 January 2019. EFM tracings from the 120 min preceding delivery were evaluated and TDA was calculated. Umbilical cord pH immediately after vacuum extraction was evaluated in correlation with TDA to evaluate TDA as a predictor of neonatal acidemia. Correlation between umbilical cord pH and neonatal birthweight was also assessed in order to evaluate the impact of neonatal birthweight on neonatal acidemia during VE.

Multiple gestations and pregnancies with known fetal chromosomal or structural anomalies were excluded. Only cases in which VE was performed due to NRFHR and when the physician suspected neonatal acidemia and umbilical cord pH was obtained were included. VE performed in the scenario of reassuring EFM (prolonged second stage, maternal exhaustion or background morbidities requiring shortening the second stage) were excluded. VE with non-continuous EFM patterns not allowing reliable calculations and VE without umbilical cord pH values were also excluded. Of note, fetal blood samplings during labor are not performed in our institution.

VEs were performed by a senior physician in accordance with the ACOG guidelines [27]. Either a Ventouse-Mityvac or Kiwi OmniCup vacuum cup was used.

The electronic medical records were reviewed, and basic maternal, labor and delivery characteristics, as well as obstetric outcomes, were extracted. EFMs in the 120 min preceding delivery were interpreted by two obstetricians who were blinded to the neonatal outcomes, as well as to their respective interpretations of EFM. 

Area under the curve (AUC) of each deceleration during the 120 min preceding delivery was calculated as follows: duration × depth × ½. TDA was calculated as the sum of AUC for each deceleration = (heart rate change in BPM*time in seconds)/2. Mean TDA was calculated based on measurements of two obstetricians that evaluated each EFM. In cases of high interobserver variability between measurements (defined as >30% difference in TDA), a third measurement was performed by a senior obstetrician and was used for the analysis. Additional EFM patterns were also retrieved, including variability in the 30 min preceding delivery (normal: amplitude range, 6–25 bpm; decreased: amplitude range, <5 bpm), [28] as well as end-stage bradycardia/tachycardia during the 10 min preceding delivery. 

Neonatal birthweights at each VE were collected and classified as low birthweight (<2500 g, LBW), normal birthweight (2500–3999 g, NBW) and macrosomia (>4000 g) [29,30].

The primary outcomes were the correlation between TDA and neonatal cord pH, and the correlation between LBW and neonatal cord pH. The secondary outcomes were the composite mild neonatal adverse outcome including one or more of the following: hypothermia, hypoglycemia, need for non-invasive ventilation, meconium aspiration and phototherapy, as well as the severe composite neonatal adverse outcome including one or more of the following: sepsis, cerebral hemorrhage, respiratory distress, anemia/blood transfusion, hypoxic–ischemic encephalopathy, convulsions and intrapartum fetal demise.

### 2.1. Data Collection

Resembling previous studies of our research group, [31] data were retrieved from the electronic medical records of the obstetric triage unit and the delivery room, which were cross-tabulated with data from the neonatal unit. Medical records were reviewed by the principal investigator to complete missing data. 

The following was collected: Maternal demographics information and medical history: age, body mass index (BMI, kg/m^2^), smoking, pregestational diabetes mellitus (PGDM), gestational diabetes mellitus (GDM) [32], chronic hypertension, preeclampsia or gestational hypertension diagnosed according to international guidelines [33].Ultrasound at presentation: placental location, polyhydramnios or oligohydramnios [34].Delivery characteristics: gestational age at delivery, use of epidural anesthesia, intrapartum maternal fever (≥38 °C during birth until 24 h after birth), amniotic fluid color, fetal sex, fetal head station and position during VE, vacuum cup type, duration of procedure, cup detachments during VE and intrapartum maternal blood loss (objectively calculated by a gravimetric machine) [35].Intrapartum EFM characteristics: TDA during the 120 min preceding delivery, variability in the 30 min preceding delivery, fetal heart rate in the 30 min preceding delivery and bradycardia/tachycardia during the 10 min preceding delivery.Neonatal outcomes: umbilical cord around body or neck, true knot in umbilical cord, Apgar scores at 5 min, umbilical cord pH, neonatal birthweights (LBW, NBW and macrosomia), Neonatal Intensive Care Unit (NICU) hospitalization rates, hypoglycemia (blood glucose < 40 mg/dL), hypothermia, need for phototherapy, respiratory distress or need for ventilation, sepsis (positive blood, urine or cerebrospinal fluid culture), cerebral hemorrhage, anemia or need for blood transfusion, hypoxic–ischemic encephalopathy, convulsions and intrapartum death.

### 2.2. Statistical Analysis

Categorical data were presented as proportions. The distribution of continuous variables was evaluated using the Shapiro–Wilk test. Continuous variables were presented by mean + SD for normally distributed variables or by median and quartiles when the distribution was not normal. Multivariable linear regression analysis was performed to evaluate factors associated with decreased umbilical cord pH. Factors included in this analysis were independent variables that were assumed to be associated with changes in umbilical cord pH based on current data. Multivariable logistic regression was performed to evaluate factors associated with the composite neonatal adverse outcome. A probability value of <0.05 was considered significant. All analyses were performed using SPSS-28 software (IBM Corp., Armonk, NY, USA). Cases of missing data (non-continuous EFM patterns not allowing reliable calculations and VE without umbilical cord pH values) were excluded.

## 3. Results

During the study period, 41,052 women gave birth in our institution, of which 3,456 had a VE. Eighty-five women met the study inclusion criteria and were included in the analysis (Table 1). Table 2, Table 3 and Table 4 present the maternal, labor and delivery characteristics of the study cohort. Median TDA in the 120 min preceding delivery was 57,725 (25th percentile, 41,060; 75th percentile, 79,965), median TDA during the 60 min preceding delivery was 39,510 (25th percentile, 29,242; 75th percentile, 55,825), median umbilical cord pH was 7.24 (25th percentile, 6.07; 75th percentile, 7.30). The cohort included 6 (7.1%) LBW neonates, 77 (90.6%) NBW neonates and 2 (2.4%) macrosomic neonates.

### 3.1. The Association between Total Deceleration Area and Umbilical Cord pH

Multivariable linear regression, adjusted for gestational age, nulliparity, diabetes mellitus and neonatal birthweight, revealed a negative correlation between TDA in the 60 min preceding delivery and umbilical cord pH. For every 10 K increase in TDA, the cord pH decreased by 0.02 (*p* = 0.038; 95% confidence interval (CI), −0.05 to −0.00). End-stage bradycardia/tachycardia, decreased variability in the 30 min preceding delivery and TDA in the 120 min preceding delivery were not correlated with a change in umbilical cord pH. Use of the Ventouse-Mityvac cup was associated with a 0.08 decrease in umbilical cord pH as compared to the Kiwi OmniCup (95%CI, −0.16 to −0.00; *p* = 0.049) (Table 5).

### 3.2. The Association between Neonatal Birthweight and Umbilical Cord pH

Performing a multivariable linear regression, as elaborated in Table 5, revealed that LBW neonates, compared to NBW neonates were not associated with a change in cord pH. 

### 3.3. Factors Associated with Neonatal Adverse Outcomes

Multivariable logistic regression analysis, adjusted for gestational age, nulliparity, diabetes mellitus and vacuum cup type, revealed that TDA in the 60 min preceding delivery was an independent risk factor for the mild composite neonatal adverse outcome (adjusted odds ratio (OR), 2.22; CI, 1.15–4.26). LBW, compared to NBW, was not associated with that outcome, as well as end-stage bradycardia/tachycardia and decreased variability in the 30 min preceding delivery.

TDA and LBW were not associated with the severe composite neonatal outcome.

## 4. Discussion

The main findings of this study are a negative correlation between TDA during the 60 min preceding VE for the indication of NRFHR and umbilical cord pH, as well as an independent association between an increase in TDA and the composite neonatal adverse outcome. The Ventouse-Mityvac cup was found to be associated with a decrease in umbilical cord pH. No correlation was found between LBW and umbilical cord pH.

The well-established terminology classifying fetal heart rate (FHR) decelerations into three categories (early, late and variable) is grounded on the belief that the mechanism behind decelerations involves a vagal response triggered by a mechanoreceptor response to fetal head compression, activation of the baroreflex due to increased blood pressure during umbilical cord compression and a Bezold–Jarisch reflex response to reduced venous return from the placenta. However, this theory ignores the natural occurrence of intrapartum decelerations. Growing evidence suggests that brief FHR decelerations during labor are a normal phenomenon, mediated by the peripheral chemoreflex in response to brief yet acute asphyxia during uterine contractions [36]. The extent to which a fetus is at risk of developing hypotension and profound acidosis depends on how well it can adapt to repeated episodes of hypoxia and the amount of time it has to recover between these episodes. Thus, the severity of the fetal oxygen deficit is linked to the depth, duration and frequency of intrapartum decelerations rather than to their timing (early, late or variable). The frequency of decelerations is crucial, as it determines the reperfusion time between asphyxial insults. This might explain why TDA has been found to be the most reliable predictor of neonatal acidemia to date [2,4].

This study aimed to determine whether TDA is a reliable predictor of neonatal acidemia in the setting of operative vaginal deliveries. We found a negative correlation between TDA and neonatal umbilical cord pH. Among the EFM features evaluated in our study, TDA was the only predictive factor of neonatal acidemia, while variability, tachycardia or bradycardia preceding delivery were not. Our findings support those of Cahill et al. [4]. However, unlike Cahill’s study, in our cohort, only TDA in the 60 min preceding delivery was predictive of acidemia. These findings are in agreement with previous reports demonstrating that TDA 30 or 60 min before delivery is the EFM feature most predictive of neonatal acidemia [37,38,39]. Additionally, since our cohort was based on vacuum extractions for patients that had an NRFHR, the 60 min prior to the vacuum-assisted deliveries has a higher impact on fetal outcome.

In support of a previous study [4], we also found TDA to be a predictor of neonatal morbidity, with an independent association between TDA and the mild composite neonatal adverse outcome. A similar association was not observed with the severe neonatal composite outcome, probably because only two cases of severe outcome occurred in our cohort. Notably, while fetal tachycardia was previously described as a contributing factor to predicting neonatal morbidity, our study did not find this association.

In spite of the previous reports regarding the association between LBW and acidemia at delivery [40], we found that LBW was not associated with a change in neonatal cord pH during VE. We assume this association was not demonstrated due to the relatively small cohort, and further investigation is required in order to evaluate whether this association exists in the scenario of VE and suspected intrapartum distress.

The Ventouse-Mityvac vacuum cup was found to be independently associated with decreased umbilical cord pH, as compared to the Kiwi OmniCup. These findings were not previously reported. Data regarding the comparison between Ventouse-Mityvac cup and Kiwi OmniCup, both non-metal, are sparse and demonstrate conflicting evidence. One study reported higher rates of neonatal head trauma with the Ventouse-Mityvac cup [22], and another did not find this association [23]. Due to the relatively small number of cases in which Ventouse-Mityvac cup was used in our cohort, as well as the undocumented reasons for choosing one vacuum cup over the other, reliable conclusions regarding its negative effect cannot be drawn, and the possible association between vacuum cup type and neonatal acidemia requires further investigation. 

### 4.1. Clinical Implications

To date, NICHD EFM categories have limited ability to predict neonatal acidemia. However, TDA seems as a promising bedside tool, replacing historic, invasive measures such as fetal scalp pH sampling [41]. Our findings further expand Cahill’s findings demonstrating that, in the context of VE performed because of suspicion of intrapartum fetal distress, an increase in TDA is associated with decreased neonatal cord pH, as well as with neonatal adverse outcomes. Our findings might also help to establish the TDA threshold at which VE should be pursued to avoid neonatal acidemia. Based on our study, low birthweight itself is not associated with lower umbilical cord pH and, thus, does not justify earlier interventions for fetal extraction, yet larger cohorts are needed in order to fully evaluate its impact. 

### 4.2. Research Implications

Nowadays, computerized tools calculating TDA using the area under the curve of each deceleration are being developed, based on electronic EFM tracings. Assuming normal neonatal cord pH values are around 7.25–7.30 [7,8], and that every 10 K increase in TDA decreases neonatal cord pH by 0.02, a >30 K increase in TDA would bring a fetus with a normal acid–base status to the range of acidemia (umbilical cord pH < 7.2) [9]. Prospective studies using computerized alarming systems should evaluate the role of TDA in labor management, warning obstetricians when TDA has reached 30 K and intervention for fetal extraction should be pursued.

### 4.3. Strengths and Limitations

The strengths of this study include the high-quality interpretation of EFM conducted by two independent physicians who were blinded to clinical and outcome data. Data were retrieved from a single medical center with consistent medical protocols, thus creating a relatively homogeneous cohort.

To our knowledge, this is the first study to evaluate TDA and neonatal birthweight as predictors of neonatal acidemia in the context of VE due to NRFHR. The main limitations of this study are its retrospective design, which dictated the exclusion of many cases due to incomplete data, resulting in a relatively small cohort and a possible selection bias including the more severe cases in which relatively higher TDAs were present and lower umbilical cord pHs were measured. This bias might impair the study’s generalizability to all VE; however, it allows reliability in VE performed when a true suspicion of neonatal acidemia is present, thus meeting the purpose of the study.

## 5. Conclusions

In VE performed for the indication of NRFHR, a correlation was found between TDA in the 60 min preceding delivery and umbilical cord pH. TDA was also found as a predictor of neonatal morbidity. Larger cohorts are needed in order to evaluate the association between neonatal birthweight and umbilical cord pH during VE.

## Figures and Tables

**Table 1 children-10-00776-t001:** Inclusion and exclusion criteria of the study cohort.

Inclusion Criteria	Exclusion Criteria
Singleton pregnancies >34 weeks GA delivered by VE	Multiple gestations
VE performed in the indication of non-reassuring fetal heart rate	Known fetal chromosomal or structural anomalies
Neonatal cord pH was obtained due to suspected fetal distress	Other modes of delivery (normal vaginal delivery and cesarean delivery)
EFM patterns allowing reliable calculations of total deceleration area	Intrauterine fetal demises
	Termination of pregnancy due to anomalies
	VE performed due to prolonged second stage, maternal exhaustion or background morbidities
	VE in which neonatal cord pH was not obtained
	Non-continuous EFM patterns
Total = 85	Total = 40,967

**Table 2 children-10-00776-t002:** Maternal characteristics of the study cohort.

Variable (*n* = 85)	n, %
Maternal age, years ± SD	30.51 ± 5.07
Gestational age, weeks ± SD	39.38 ± 1.23
Nulliparity	62 (72.9%)
Maternal BMI, kg/m^2^ ± SD	22.86 ± 4.62
Obesity (BMI > 30)	5 (10.9%)
Smoking	6 (7.1%)
Diabetes mellitus	8 (9.4%)
Hypertensive disorders	2 (2.4%)

BMI—body mass index; n—number; SD—standard deviation. Data are presented as number (rate) or mean ± standard deviation, as appropriate. Diabetes mellitus includes pregestational and gestational diabetes; hypertensive disorders include preeclampsia and gestational hypertension.

**Table 3 children-10-00776-t003:** Labor and delivery characteristics of the study cohort.

Variable (*n* = 85)		n, %
Epidural anesthesia		84 (98.8%)
Intrapartum fever		10 (12.0%)
Amniotic fluid color	Clear	55 (68.8%)
	Meconium	23 (28.7%)
	Bloody	2 (2.5%)
Occiput anterior position		69 (81.1%)
Fetal head station at vacuum extraction	Mid-pelvis	47 (58.8%)
	Low	28 (35.0%)
	Outlet	5 (6.3%)
Vacuum cup type	Kiwi OmniCup	55 (67.9%)
	Ventouse-Mityvac	26 (32.1%)
Procedure duration, min ± SD		4.99 ± 3.22
Vacuum cup detachment		17 (21.0%)
Decreased variability in the last 30 min		3 (4.3%)
End-stage bradycardia		2 (2.5%)
End-stage tachycardia		16 (19.8%)
Umbilical cord around body		3 (3.6%)
Umbilical cord around neck		26 (31.0%)
True knot of umbilical cord		1 (1.2%)
Umbilical cord pH ± SD		7.19 ± 0.14
Apgar 5 min < 7		1 (1.2%)
Neonatal sex	Male	55 (64.7%)
	Female	30 (35.3%)
Neonatal weight, g ± SD		3197 ± 435
Low birthweight (<2500 g)		6 (7.1%)
Normal birthweight (2500–3999 g)		77 (90.6%)
Macrosomia (≥4000 g)		2 (2.4%)
NICU admission		1 (1.2%)
Mild composite neonatal outcome *		41 (56.2%)
Severe composite neonatal outcome **		2 (2.4%)
Maternal blood loss during birth, mL ± SD		352 ± 256

n—number; SD—standard deviation. Data are presented as number (rate) or mean ± standard deviation, as appropriate. * Including hypothermia, hypoglycemia, need for non-invasive ventilation meconium aspiration and phototherapy. ** Including sepsis, respiratory distress, cerebral hemorrhage, anemia/blood transfusion, hypoxic–ischemic encephalopathy/convulsions and intrapartum death.

**Table 4 children-10-00776-t004:** Delivery characteristics—continuous parameters.

Variable	Median	25th Percentile	75th Percentile
Third-stage duration (h)	0.16	0.08	0.37
Second-stage duration (h)	2.22	0.68	0.18
Total deceleration area 60 min before delivery *	39,510	29,242	55,825
Total deceleration area 120 min before delivery *	57,725	41,060	79,965
Umbilical cord pH	7.24	7.07	7.30
Base excess	−6.80	−9.20	−4.70
Blood loss during delivery (mL)	300	200	400

* Expressed as (heart rate change in BPM × time in seconds)/2.

**Table 5 children-10-00776-t005:** Multivariable linear regression, factors associated or not associated with a decrease in umbilical cord pH.

Factor	Decrease in Umbilical Cord pH	95% CI Lower Limit	95% CI Upper Limit	*p* Value
Total deceleration area in the last 60 min of delivery (every 10 K increase)	**−0.02**	**−0.05**	**−0.00**	**0.038**
Normal variability last 30 min	−0.02	−0.24	0.20	0.826
End-stage bradycardia	−0.09	−0.31	0.13	0.399
End-stage tachycardia	0.02	−0.08	0.11	0.719
Nulliparity	0.01	−0.08	0.09	0.825
Diabetes mellitus *	0.00	−0.14	0.14	0.968
Gestational age at birth (every additional week)	−0.01	−0.05	0.02	0.391
Vacuum cup Ventouse-Mityvac (compared to Kiwi OmniCup)	**−0.08**	**−0.16**	**−0.00**	**0.049**
LBW neonate (compared to NBW)	−0.09	−0.23	0.09	0.328

Factors included in the multivariable linear regression model were independent variables thought to be associated with a decrease in neonatal cord pH based on current data. Data are presented as the decrease observed in umbilical cord pH and its 95% confidence intervals (CIs). * Diabetes mellitus includes pregestational and gestational diabetes.

## Data Availability

The data presented in this study are available upon request from the corresponding author.

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
