# Peer review of "The Impact of Total Deceleration Area and Fetal Growth on Neonatal Acidemia in Vacuum Extraction Deliveries"

_children, 2023, doi:10.3390/children10050776_

Round 1
Reviewer 1 Report
I must congratulate the authors!
It s been a long time since I had the opportunity to read such interesting research and such a well-written paper.
The statistical correlations are very sound from a scientific point of view.
I am a neonatologist and I've seen lots of vacuum extractions but I never wondered about the correlations made by the authors of this paper.
Only fine minor spelling issues found in this paper
Author Response
Reviewer 1
Comments and Suggestions for Authors
I must congratulate the authors!
It s been a long time since I had the opportunity to read such interesting research and such a well-written paper.
The statistical correlations are very sound from a scientific point of view.
I am a neonatologist and I've seen lots of vacuum extractions but I never wondered about the correlations made by the authors of this paper.
Comments on the Quality of English Language: Only fine minor spelling issues found in this paper
Response:
Thank you for your review, we are glad to hear you found our manuscript interesting. Our manuscript has been revised, correcting all spelling issues found.
Reviewer 2 Report
In this study a correlation between total deceleration area and neonatal acidemia along with neonatal birthweight is analysed on a sample of 85 vacuum extractions. The authors report a negative correlation between TDA and umbilical pH. I have the following comments:
1. In the introduction section please explain the abbreviation NICHD
2. Do the authors use only Kiwi and Mityvac vacuum extractors in their institutions or is a metal cup vacuum also used? A short explanation on the most commonly used vacuum extractors should be added in the introduction section along with failure rates of various vacuum extractors.
3. The inclusion and exclusion criteria should be presented in a table. Is fetal blood sampling performed in the authors’ institution during labour? If so, were these patients excluded?
4. The two obstetricians who interpreted EFM were blinded to the neonatal outcomes. Were they also blinded to their respective interpretations of EFM?
5. In the discussion section the authors should describe how they would set up a prospective study to evaluate the clinical usefulness of TDA calculation. Would it be feasible to calculate it during labour?
Reviewer 3 Report
Dear Authors,
I would like to ask for changing the way of naming birth weight into more popular way – line 19 and line 91 in Materials and Methods:
Low birth weight, normal birth weight...like in the article:
https://www.ncbi.nlm.nih.gov/pmc/articles/PMC9287292/
I would suggest to describe more the pH value and its role in detecting the acidosis. Maybe the article https://pubmed.ncbi.nlm.nih.gov/34567841/ would be helpful.
Round 2
Reviewer 2 Report
I have no further comments